# A Novel Cathode Material Synthesis and Thermal Characterization of (1-x-y) LiCo_1/3_Ti_1/3_Fe_1/3_PO_4_, xLi_2_MnPO_4_, yLiFePO_4_ Composites for Lithium-Ion Batteries (LIBs)

**DOI:** 10.3390/molecules27238486

**Published:** 2022-12-02

**Authors:** Lu Li, Xin Min, Majid Monajjemi

**Affiliations:** 1State Key Laboratory of Electrical Insulation and Power Equipment, Center of Nanomaterials for Renewable Energy, School of Electrical Engineering, Xi’an Jiaotong University, Xi’an 710049, China; 2School of Environmental Science and Engineering, Shanghai Jiao Tong University, Shanghai 200240, China; 3Department of Chemical Engineering, Central Tehran Branch, Islamic Azad University, Tehran 1496969191, Iran

**Keywords:** lithium-ion battery, various cathode material, LiCoO_2_ toxic, sol–gel method, thermal characterization

## Abstract

Lithium-ion batteries are known for their high efficiency for storing electrical energy, especially for hybrid vehicles. In this research, the development of mixture composites in the cathode electrode of LIBs has been discussed and designed based on ternary solid solutions. We have given a novel synthesis and method preparation of cathode electrode materials to reduce costs while increasing the efficiency and simultaneity for the future of these technologies. The major problem in the LIBs is related to LiCoO_2_ as a popular cathode material that, although it has a high efficiency, is expensive and very toxic. Therefore, the usage of a lower weight of cobalt compared to the LiCoO_2_ cathode material is economically advantageous for this research. Several samples of the (1-x-y) LiCo_1/3_Ti_1/3_Fe_1/3_PO_4_ xLi_2_MnPO_4_ and yLiFePO_4_ system were synthesized via sol–gel experiments. Various stoichiometric amounts of the LiNO_3_, Li_2_MnPO_4_, Mn (Ac)_2_. 4H_2_O, Co (Ac)_2_.4H_2_O, Ti(NO_3_)_2_.6H_2_O and LiFePO_4_ have been used for several compositions of chrome, manganese, cobalt and titanium in 28 samples of (1-x-y) LiCo_1/3_Ti_1/3_Fe_1/3_PO_4_. By using thermal characterization, five samples have been selected due to their conditions in viewpoints of capacity and cyclability as well as activation energy, which is one of the major factors. These composites exhibited fairly consistent charge/discharge curves during the electrochemical testing. From the viewpoint of the physical and chemical properties, among these samples, the Li_1.501_Co_0.389_Ti_0.055_Fe_0.055_Mn_0.501_PO_4_ structure has a high efficiency compared to other compositions.

## 1. Introduction

### 1.1. Lithium-Ion Batteries Background

In the past two decades, Goodenough et al. used LiFePO_4_ as cathode electrode materials for lithium-ion secondary batteries [1], which include 3.5 g/cm^3^ density and a theoretical capacity of 175 mAh/g [2]. LiFePO_4_ has many advantages, such as a lengthy cycle life, security and safety, suitable power densities and, most notably, low cost. Strong voltage and life cycles without toxic [3,4] materials inside rechargeable batteries such as LIBs are vital items for the management of important systems. It must be emphasized that three major items in LIBs must always be analyzed for controlling the charge–discharge process in each battery: voltage, current and temperature. Since the LiFePO_4_ has adopted the olivine structure—and also due to its stability, different charge, quick cut-off voltages and fast production process—it has been selected for further testing and discussion for its contributions with the other metals mentioned, especially cobalt. In the earth, LiFePO_4_ compounds exist in lithium iron phosphate ore from the reaction as follows: in charging: $O_{4}-xLi^{+}-xe^{-}\to xFePO_{4}+\left(1-x\right)LiFePO_{4}$ and in discharging:

$FePO_{4}+xe^{-}+xLi^{+}\to xLiFePO_{4}+\left(1-x\right)FePO_{4}$. The charge/discharge reaction happened between the LiFePO_4_ and FePO_4_ phases. In other words, in the charging situation, Li^+^ exits from LiFePO_4_ and makes the FePO_4_ phase. In the discharging time, Li^+^ with FePO_4_ makes the LiFePO_4_ phase. By any delay in charge, the internal resistance and voltage will increase quickly with Li^+^ decreasing. It is notable that the charge process is accomplished to be more difficult than the discharge under low temperatures, and the usual solution for removing this problem is only to charge the battery in high temperature relatively. In a slow voltage, LIBs have a stable internal resistance, therefore the heat rises slowly and the decreasing life cycle causes problems for charging that requires more energy. The third row of elements in the Mendeleev table, including Mn, Ti and Co (MTC), are important elements for LIBs due to their properties for high-voltage operations and large energy storing in a little space. Titanium is unstable, and has a propensity towards destructive reactions with the electrolyte. A major problem of Co is its cost, which is approximately several times more costly than the Ti [5,6]. The safety factor is also an important item for LIBs that appear only in large-sized batteries (Table 1). The biggest advantages of these types of composites with mixing metal element materials are their ability to prevent the explosion of batteries that can occur anywhere and anytime. The goals of this work are based on preparing a ternary solid solution for cathode material for decreasing the amount of cobalt usage in LiCoO_2_, which is expensive and toxic, and whether it will possible to control the explosion of LIBs [7,8,9].

Several works about LIBs were accomplished by Rossen [10] and Saphr et al. [11], who indicate Ti and Mn are active ions that can be presented in “+2” and “+4” oxidation rather than “+3”. Titanium with forms between +2 and +4 valences situations is an active element and also its salts suitable for cathode materials; meanwhile manganese is +4, which remains without Jahn-Teller distortion with the +3 valance. Makimura and coworkers [12] presented an initial discharging around 155.mAh/g for 35 cycles at 4.5 V and their results exhibited 200 mAhg^−1^ rechargeable capacities for LiTi_1/2_Mn_1/2_O_2_ among the voltages of 2.2–4.6 V after 35 cycles. Our work is based on mixing proper mole fractions of various transition elements that yields an advantageous performance; therefore, these materials are derived from Li (Ti (1-X-Y) Mnx Coy) O_2_ categories that were first published in 1999 and 2000 by Liu and Yoshio [13,14]. In this work, we selected a combination of LiCo_1/3_Ti_1/3_Fe_1/3_PO_4_. For appropriate stoichiometry of x = 1/3, for each iron of cobalt and titanium due to +3 oxidation numbers, such as Mn^+4^, Co^+3^ and Ti^+2^ in the ranges between 2.0 up to 3.9 volte, lithium-ions can be transferred easily between two redox reactions of Co^+3^/Co^+4^ and Ti^+2^/Ti^+4^ in this situation. In addition, LiTi_0.4_Mn_0.4_Co_0.2_O_2_, [15,16,17] Li (Ti_0.8_Co_0.2_) O_2_ and LiTi_0.8_Co_0.15_Al_0.05_O_2_ cathode materials [18] have been selected for rating the capabilities of these composites, which have smaller thermal stabilities than LiCoO_2_.

Li_2_MnPO_4_ which can be demonstrated as Li [Li_1/3_Mn_2/3_] PO_4_, has a similar structure to LiCoO_2_ and contains a super-lattice ordering of Li+ and Mn^4+^ in transition ions shells. This compound consists of lithium and manganese ions in alternating (1:2 ratio respectively) positions that are separated by cubic oxygen. Since oxidization numbers more than +4 for Mn^4+^ are impossible in low voltages, for any further oxidization, a number of +4 are needed for high voltages; therefore, this compound might be electrochemically inactive in low voltages. Improving the capacity of advanced lithium (Li) batteries can be improved by enhanced electronic conductivities and mixing cathode materials such as various composites. Cathodes in these battery cells may be comprised of metal salts of relatively low electronic conductivity, and separator/electrolyte compositions must be tuned to readily admit ions. In addition, Electrolyte engineering improved cycling of Li metal batteries and anode-free cells at low current densities; however, the high-rate capability and tuning of ionic conduction in electrolytes are favorable [19]. The challenges faced by researchers in this field include the relatively low electrical and ionic conductivity values in the cathode. Models of battery cells and materials widely require the best available estimates for conductivity and diffusivity in order to both predict responses and design improved composite materials. Therefore, solid-state electrolytes with sufficient ionic conduction, wide voltage windows, flexible-rigid interfaces and ease of manufacturing are considerable to the development of solid-state lithium metal batteries [20]. However, experimental validation of such parameters is often challenging. Often, the characterization of battery materials is carried out as a postmortem analysis. Experimental setup is invaluable for interpreting results because they can only describe the end state. A key challenge is developing an experimental technique for cathode materials to detect Li-ion movement within the electrode lattice structure and the electrode–electrolyte interface as a function of time [20,21]. Our objectives in this work are to find a suitable cathode material based on electrical and ionic conduction theories for estimating a suitable composite with a mixing of three cathode materials: LiCo_1/3_Ti_1/3_Fe_1/3_PO4, LiMn_2_O_4_, and LiFePO4.

### 1.2. Entropy Changing and Gibbs Energy in LIBs

The free potential of an electrochemical cell can be generated via testing compared with the theoretical potential. Both theoretical and free potential can be simulated by the type of cathode materials and the electrolyte contained in the LIBs [22]. In a real cell, it is favorable that all of the Gibbs energies transform to advantageous electrical energies in time of discharge. Sometimes, the energies are wasted due to polarization mechanisms via loading current inside the electrochemical reactions and converted to heat. In LIBs, the cathode material is spongy, and lithium-ions can be diffused in electrolytes during the charging/discharging [23,24,25,26]. Therefore, the polarizations are more complicated and the heat generation in the electrolytes is also more intricate during any battery process; consequently, the Gibbs energy in LIBs is high. Heat in the LIBs can be produced because of entropy changes due to cathode materials with electrolyte reactions, as well as ohmic heating due to the Joule’s effect. Moreover, heat can be endothermic for the charging process and exothermic for the discharging duration [22,27]. This can be calculated through equation:(1)$$Q=I\left(V_{oc}-V\right)-I[T\left(\frac{dV_{oc}}{dT}\right)]$$
where $V_{oc}$ is the voltage of the open circuit and V is the cell voltage. In Equation (1), the first phrase is the generated heat by ohmic effects (irreversible phenomenon) and the second term is the generated heat due to the reversible entropy changing in the cell [28]. It is notable that in a hybrid electric vehicle (HEV), the second phrase is normally negligible compared to the first term [29,30,31,32,33]. As heat is produced during the charging/discharging phenomenon, the necessity for sustainable cooling and battery thermal management is needed and can be controlled through insulation, liquid systems and phase-change materials [34,35]. Pals et al. [36,37,38] developed a thermal system of an Li|PEO_15_–LiCF_3_SO_3_|TiS_2_ electrode pair for the single cell of LIBs for energy balancing using Equation (2):(2)$$Q=I\left(V_{oc}-V-T\frac{\partial V_{oc}}{\partial T}\right)=h_{conv}A_{cell}\left(T-T_{\infty }\right)+m_{c}c_{p}\frac{dT}{dt}$$

The left side of Equation (1) exhibits the heat generation terms. Several simulated systems were accomplished with both isothermal and adiabatic boundary positions for investigating the performance of LIBs. It is important to determine the state of charges (SOC) during the next process. At each SOC, the cell temperature decreased from 25 °C to 15 °C and the open circuit potential (OCP) was monitored. After completing the last temperature step at 10 °C, the temperature rises to 25 °C, and then a 10% amount is applied to the SOC. The entropy and enthalpy can be calculated based on Equations (1) and (2), respectively, as follows [38]: (3)$$\Delta S_{\left(\chi \right)}=F{\left(\frac{\partial E_{0}\left(\chi ,T\right)}{\partial T}\right)}_{\chi ,P}$$and
(4)$$\Delta H_{\left(\chi \right)}=-F[E_{0}\left(\chi ,T\right)+T{\left(\frac{\partial E_{0}\left(\chi ,T\right)}{\partial T}\right)}_{\chi ,P}]$$
where $E_{0}\left(\chi ,T\right)$ is the OCP at temperature T, χ=SOC, F is the Faraday constant and p = pressure. Nelson et al. [39] used a cumulated model to explore the ability of Li-ion hybrid-electric vehicle (HEV) designs of several geometries to evaluate thermal requirements. Smith et al. [40] extended a developed 1D electrochemical phenomenon. By using this model, the physicochemical property values were temperature dependent based on the Arrhenius equation, which defines the temperature sensitivity of a general physiochemical property, Ψ, as follows:(5)$$\Psi =\Psi_{ref}\exp[\frac{E_{act}^{\Psi }}{R}(\frac{1}{T_{ref}}-\frac{1}{T})$$
where $\Psi_{ref}$ is the property value defined at reference temperature and $T_{ref}$ = 25 °C. Here, the $E_{act}^{\Psi }$ controls the temperature sensitivity [41,42,43]. In viewpoint of temperature dependence, we define operating maps of discharge and charge power capability as a function of temperature initial condition and cell temperature.

## 2. Results

### 2.1. Material Characterization

#### 2.1.1. SEM Analyzing

These tests have been accomplished for confirming the composition for evaluating the morphology of all synthesized composites in Table 2. These SEM morphologies have been presented in proper phases (1–5 microns for particle sizes) that are important for increasing the performance of the cathode materials (Figure 1, Figure 2 and Figure 3).

#### 2.1.2. X-ray Diffraction

X-ray diffraction was accomplished for the 28 samples as well as those that were repeated to distinguish whether the correct phase was achieved or not. X-ray photoelectron spectroscopy (Nexsa G2 XPS) and inductively coupled plasma atomic emission spectroscopy (ICP-AES) characterization techniques were only done on the samples that showed proper results and were repeated a few times. XPS was accomplished to have a precise analysis about the percentage amount and the oxidation states of the elements present in the composition. ICP was performed for the detection of iron, titanium, manganese and cobalt and also to verify the exact composition of the sample. X-ray diffraction was done on 28 items using a powder X-ray diffraction system that utilized Cu Kα radiation with λ = 1.55 Å (Figure 4 and Figure 5).

The θ angles in the X-rays, based on Bragg’s law, will be incompatible with the composites and sharp peaks observed. These sharp peaks are used to analyze the structure of the 28 samples and lattice parameters and determine the formation of a pure phase to recognize the existence of any unknown substances by comparing the observed diffraction data against a database maintained by the International Centre for Diffraction Data (Figure 5 and Table 3).

In addition, the inductively coupled plasma atomic emission spectroscopy (ICP-AES) technique has been used to identify transition metals in the compositions of 28 samples.

ICP analysis was accomplished on 28 samples by a PerkinElmer Optima 7300 DV with a Cetac ASX- 520 auto sampler, Mein hard concentric nebulizer and a cyclonic spray chamber with baffle. In addition, a statistical analysis was done to find the most proper materials and to explore the trends in the capacity and cycle ability range over the 28 items of the composition triangle.

#### 2.1.3. Charge and Discharge Measurement

The charge/discharge was measured by BTS-610 battery tester with charge / discharge rates between 0.1 C to 0.4 C; 1 C is equal to 170 mA g^−1^. During the testing, the cell was activated at 25 °C by five charge/discharge cycles as follows: charge of 4.5 V and then discharge of 2.2 V, then it was soaked at 25° for 10 h to achieve thermal equilibrium. In the second step, the test was started as follows: charge of 4.5 V at 0.1 C, Charge of 4.2 V for 3 h, and finally discharge of 2.0 V. After an electrochemical testing impedance spectroscopy measurement has been done using electrochemical impedance spectroscopy (EIS) including the electrochemical analyzer (CHI604b), the cells were dissembled and a three-electrode system was assembled in a glove box. The LiFePO4 electrode was used as the working electrode, and lithium foils were used as the counter and reference electrodes. The signal of AC voltage of ±5 mV was used in a frequency range between 10^−2^ and 10^5^ Hz. The obtained impedance was fitted by ZsimpWin300 Program.

### 2.2. Thermal Conductivity Measurement

To determine thermal conductivity, two temperature gradients were forced across the thickness in two side directions of the cell based on Aiello’s work [39]. Temperature amounts were measured both on the hot and cold sides of the cell via K-type thermocouples. The heat flux (Hukse flux Thermal Sensors, Delft, and The Netherlands) is calculated with a measurement accuracy of 0.5%. The thermal conductivity can be calculated by inserting the measured values into the K = (F × d)/∆T Equation, where F (W/m^2^) describes the determined average heat flux, ∆T(k) is the forced temperature gradient and d is the average thickness of the sample [39].

## 3. Discussion

### 3.1. Cathode Material Compositions

This investigation accomplished finding the best cathode material compositions including (1-x-y) LiFe1/3Ti1/3Co1/3PO4 xLi_2_MnPO_4_ and yLiFePO4 mixtures with high initial discharge capacities, suitable cyclability and inexpensive costs compared to traditional LIBs. Therefore, 28 composite points based on the lever rule have been selected (Figure 1). In addition, stoichiometric weights and mole-fractions were provided to find an optimized material with good electrochemical efficiency. These composites were determined through the triangle diagram (Figure 1 and Table 2, Table 3 and Table 4) and synthesis by the sol–gel method. Ti and Co amounts decreased towards the down direction of the triangle, meanwhile the compositions of 22 and 24 have zero Ti and Co percentage. High Mn value is found in sample 22 and its content decreases at the opposite end points of the triangle. High Fe value is found in sample 28 and its content decreases towards the top and at the opposite end points of the triangle. Cobalt percentage is found in a wide region in the triangle and also decreases near Li (Li_0.33_Mn_0.66_) O_2_. Since the capacities and cyclability of the compositions are directly related to the amounts of Mn, Co, Ti and Fe, specific capacities for all samples must be determined as the amounts of energies which can be reserved in volume or mass (Ah). Although the rate capability is related to their design and varies considerably between different manufacturers, they can be determined as the rate at which the cell is being charged. The C-rate and the discharge capacities have been compared with 18650-type “C/LiCoO_2_ Sony battery” results and the samples were tested via a cycler (Arbin BT 2000 battery testing system), between 2.4 V and 4.6 V with low C-rate of C/12. The initial discharge capacities varied from 102 mAh/g to 248 “mAh/g”. Both capacity and cyclability increase from LiFe_0.333_Ti_0.333_Co_0.333_PO_4_ towards the binary composition of Li_2_MnO_3_ and LiCoO_2_. Although sample 20 shows the high capacity of “248.1” mAhg^−1^, it contains a low cyclability. Sample 18 exhibits a good capacity of 220.2 mAh/g with a high cyclability (95). Meanwhile, due to high amount of Mn^4+^ ion in sample 22, it has a low capacity. Although these kinds of data are not sufficient for determining a suitable cathode material in the viewpoint of capacity and cyclability amounts, the statistical analysis can be useful for finding the region of the best item from the data of the 28 compositions. Therefore, any testing with both capacity and cyclability relation in viewpoint of the triangle regions is needed. In this work, the SATISTICA software has been used for analyzing the data; X, Y and Z are Li_2_MnO_3_, Li_2_CoO_2_ and LiFe_0.333_Ti_0.333_Co_0.333_PO_4_, respectively, including two diagrams of the capacity and cyclability (Table 4 and Figure 6 and Figure 7). 

Although the structure of Li_2_MnO_3_ or Li [Li_1/3_Mn_2/3_] O_2_ is similar to that of LiCoO_2_, there is a Mn^4+^ and Li^+^ super lattice sequence in the transition layers. Li_1.501_Fe_0.389_Ti_0.055_Co_0.055_Mn_0.501_PO_4_ can be shown as a solid solution with Ti doped on the Fe side between Li_2_MnPO_4_ and LiFePO_4_ and can be a promising cathode material due to its improved stability and electrochemical performances. According to the analysis made among these compounds, three samples were selected, synthesized, characterized and tested with high sensitivity. According to the initial discharge, capacity and some extra result numbers, Figure 1, Figure 2 and Figure 3 were selected and stated as the suitable cathode materials among these structures.

### 3.2. XRD & XPS

An X-ray photoelectron spectroscopy (XPS) instrument accomplished the testing of 28 samples for determining the valence states of transition metals containing Fe, Ti, Mn and Co. Figure 8 shows the XPS scans for sample 18.

For Ti, the binding energy of the main peak was found to be at 952 eV, corresponding to Ti 2p^3/2^ core level, which is indicative of Ti^+2^. XPS results showed the valence state of Ti synthesized in sample 18 is +2. Manganese 2p core levels in which the 3/2 and 1/2 orbit doublet components perfectly suggest that Manganese can be presented with an oxidation state of +4. While Cobalt 2p core levels spectra showed peaks corresponding to Cobalt 2p^3/2^ (890 eV) indicative of Co+3, as in LiCoO_2_, and are low spin in nature. Therefore, the predominant valence states of the transition metals are determined to be Ti^+2^, Co^+3^ and Mn^+4^. This result implies that the primary redox couple involved in the chemical reaction process will be Ti^+2^/Ti^+4^, Co^+3^/Co^+4^ and because Mn is already presented in its maximum oxidation state of +4, it is not oxidized in this voltage. Figure 8 shows the XPS spectra of all remaining compositions showing the electronic transitions of the different elements presented.

### 3.3. Electrochemical Testing

Since sample (c) (Figure 9) is pure material, sample (a) “Li_1.501_Fe_0.389_Ti_0.055_Co_0.055_Mn_0.501_PO_4_” is suggested as the best composition for the cathode material in this study. Obviously, the lower weight of cobalt used in these samples compared to LiCoO_2_ is a cost advantage. Samples were replicated in T-Cells and electrochemically tested using the original conditions, and cycling was carried out at room temperature with a constant C ratio of C/12 from 2.4 to 4.6 V (Figure 9 and Figure 10).

All of the samples were tested with the Arbin BT 2000 battery testing system or cycler. They were cycled between 2.4 V and 4.6 V, and a constant current of 300 µA was supplied. A C-rate of C/12 was maintained for all of the cells. A minimum of four cycles were performed for all the cells to analyze the cyclability of the sample. The voltage and current are programmed using the MITS software available in the computer attached to the cycler.

It was found that all samples of 16, 23, 10, 18, 25, 27, 8, 5 and 28 composites were synthesized by the same method with the best conditions in terms of capacity and recyclability. Sample 18 (Figure 9) with Li_1.501_Fe_0.389_Ti_0.055_Co_0.055_Mn_0.501_PO_4_ structure and minimum energy activation is the best composition for the Ti-doped position because sample 28 is pure and samples 27 and 25 are binary compositions from those samples. The samples were electrochemically tested in T-Cells using original conditions and cycling at room temperature between 2.4–4.6 V with a constant C ratio of C/12 (the repeated sample 18 was made of T-Cells and subjected to electrochemical testing using the original conditions and the cycling was accomplished between 2.4–4.6 V with a constant C-rate of C/12 at room temperature) and it was found to have better charge and discharge capacities (155, mAhg^−1^) in this study [36,38].

### 3.4. Thermo Dynamic Properties Calculation

In this work, we present a purely experimental thermal characterization of thermo-dynamic properties (Table 4) and operating behavior of a lithium-ion battery utilizing a promising electrode material, LiFePO_4_, using Equations (1)–(4). The thermal conductivities of the LiFePO_4_ positive electrode and negative electrode materials were found to be 1.8 ± 0.2 W/m°C and 1.2 ± 0. 2 W/m°C, respectively [34,35]. Based on thermal characterization data and energy activation, low temperature performance of the LiFePO_4_ cathode is investigated by charge/discharge tests and electrochemical impedance spectroscopy (EIS). The results show that the effect of charge temperature on charge and discharge capacity are different for any kinds of the 28 compositions within a very slight range of −25 to 30 °C, and the influence of discharge temperatures on discharge capacities also have larger amounts. Although these kinds of systems can help remove the disadvantages of cobalt, which is expensive and toxic, the performances of these kinds of systems are similar to the LiCoO_2_ cathode material [32,33]. Finally, the thermodynamic parameters have confirmed the stability and high efficiency of these systems of cathode materials.

Electrochemical energy storage devices are conveniently characterized using the Gibbs free energy (G = H − TS) and have been calculated from entropies and enthalpies (Table 5); activation energies are estimated from the data by Botte et al [40]. Each case is run under isothermal operating conditions (Table 5).

Charge–discharge testing was accomplished to enable thermal management designing solutions. Heats generated by the battery along with surface temperature and heat flux at the distributed locations were measured. As a result, the maximum value of the total heat generated was 41.34 kJ during the same discharge conditions.

## 4. Materials and Methods

### 4.1. Cathode Electrode Preparation

The composites were synthesized using the sol–gel method due to the simple chemical reaction with a low temperature and a high degree of homogeneity. Stoichiometric weights of the LiNO_3_, Li_2_MnPO_4_, Mn (Ac)_2_ ∙4H_2_O, Co(Ac)_2_⋅4H_2_O, Ti(NO_3_)_2_⋅6H_2_O and LiFePO_4_ as derived materials of lithium, iron, manganese, cobalt and titanium in 28 samples of (1-x-y) LiFe_1/3_Ti_1/3_Co_1/3_PO_4_ have been used (Table 2 and Table 3). These mixtures were first dissolved in 50 mL of DI H_2_O and then equivalent molar weights of citric acid were added. The electrolytes were prepared by dissolving one mol LiPF_6_ in a blend of ethylene carbonate (EC) and dimethyl carbonate (DMC). Ethyl methyl carbonate (EMC) in a dry glove-box is filled with high pure argon. Solvents of EC, DMC and EMC were dried by molecular sieves for several days. Viscosity and ionic conductivity of the electrolyte solutions were measured over a wide temperature range of −45 to 2 °C. The electrodes were prepared through mixing 70 wt% LiFePO_4_/C powder, 20 wt% acetylene black and 10 wt% polyvinylidene fluoride (PVDF) in NMP, and then the slurry was coated onto Al foil current collector and dyed at 120 °C. The lithium metal foil was used as the anode and the 2025-type coin cell (Li/LiFePO_4_) was assembled in the glove box. Finally, the clear solution was slowly dried and turned into gel. This solution was continuously stirred for about 20 min for the formation of a homogeneous mixture of gel and is then kept under a hot-plate for 10–15 h around 95 °C, so all the distilled water gets evaporated.

### 4.2. State of Charge (SOC) Measurements

A thermodynamic measurement system instrument, the ETMS battery analyzer BA-1000 KVI PTE LTD, has been applied for running the conditioning cycles. Cells are charged to 4.5 V under C/12 rate then a constant 4.5 V was applied until the current dropped below 0.1 mA. Next, the cells were discharged to 2.4 V under a C/12-rate and a constant 2.4 V voltage was held until the current dropped again below 0.1 mA. In this cycle, the ETMS determines the cells’ charge and discharge capacities.

### 4.3. Applied Equipment

The phase’s identities and the crystal combination of all compositions were measured using single crystal X-Ray diffraction (SC-XRD)D8 QUEST ECO Bruker equipment from Germany) and their morphologies were analyzed through a scanning electron microscopy (SEM) using a scanning electron microscope (Hitachi S-4000, Tokyo, Japan). In the cells, Li metal (99.9%, Aldrich Chem., Milwaukee, WI, USA) was used as the anode and reference electrodes, 1 M LiPF6 in ethylene carbonate (EC)/diethylene carbonate (DEC) (1:1 volume ratio) (Tomiyama Pure Chem. Ind., Tokyo, Japan) was used as the electrolyte, Celgard 2400 membrane (Hoechst Celanese Corp., Charlotte, NC, USA) as the separator, and Li_2_MnPO4-based composites as the cathode. In the preparation of the composite cathodes, spinel powder was mixed with acetylene black (100%, Strem Chem., Newburyport, MA, USA). These tests have been accomplished for confirming the composition needed for evaluating the morphology of our synthesized composites (28 samples). These SEM morphologies have been presented in suitable phases (between 1–5 microns for particle sizes) that are important for increasing the performance of the cathode materials (Figure 1, Figure 2 and Figure 3).

## 5. Conclusions

The (1-x-y) LiFe_0.333_Ti_0.333_Co_0.333_PO_4_, xLi_2_MnPO_4_, yLiFePO_4_ composites as cathode materials, after being successfully synthesized by the sol–gel method, systematically analyzed the effects of charge/discharge capacities as well as capacity retention. The structural and electrochemical properties were investigated and examined. The results show that all prepared “Li_1.501_Fe_0.389_Ti_0.055_Co_0.055_Mn_0.501_PO_4_ structure”-type layered structures significantly improved capacity retention, regardless of titanium content and Mn contribution. It also suppresses the phase transitions that usually occur during the cycle in the i_1.501_Fe_0.389_Ti_0.055_Co_0.055_Mn_0.501_PO_4_ structure and the charge–discharge reversibility. Improvement of the composition indicates that the percentage of titanium and cobalt performs well. Such systems help to overcome the disadvantage of costly and toxic cobalt. The performance of such systems is similar to LiCoO_2_ cathode material. Therefore, it is recommended to manufacture lithium-ion batteries using several transition elements such as Mn, Al and Mg for further research.

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
