# Peer review of "A Novel Cathode Material Synthesis and Thermal Characterization of (1-x-y) LiCo1/3Ti1/3Fe1/3PO4, xLi2MnPO4, yLiFePO4 Composites for Lithium-Ion Batteries (LIBs)"

_molecules, 2022, doi:10.3390/molecules27238486_

Round 1

Reviewer 1 Report

The authors undoubtedly show insufficient background and fundamentals on electrochemistry and rechargeable batteries, which results in misleading results and judgements. Major revision should be done before reconsideration:

(1) All the figures should be redrawed according to the appropriate format instead of simply capturing from the analysis software, such as XRD tests.

(2) The electrochemical performances of different cathodes are misleading. The major concerns of battery performance are missing, such as cycling stablity, rate performance, high-rate stability, coulombic efficiency, impedance and so forth. Please refer to publications of lithium-ion batteries and find the correspinding evaluation. 

(3) In my opinion, the major concerns of lithium-ion batteries associate with the insufficient energy density, poor stability and safety issues, and the LiCoO2 remains the major cathode that has applied in 3C electronics.

(4) The paper is not well-prepared, as many of tests are involved without any results rather than the disscussion.

Author Response

Dear Editor

Many thanks for your cooperation and referee comments which are pretty helpful for improving the manuscript. In the following there are the responses to the reviewer (point by point)

                                                 reviewer1#

The authors undoubtedly show insufficient background and fundamentals on electrochemistry and rechargeable batteries, which results in misleading results and judgements. Major revision should be done before reconsideration:

(1): The electrochemical performances of different cathodes are misleading. The major concerns of battery performance are missing, such as cycling stablity, rate performance, high-rate stability, coulombic efficiency, impedance and so forth. Please refer to publications of lithium-ion batteries and find the correspinding evaluation. 

Response to (1): I agree with the referee , Although I did as possible the referee comments , it must be considered that we are working with 28 samples that is a large number and the comparison of these large number for finding the best composition of   testing, relatively is our goal. 

(2): In my opinion, the major concerns of lithium-ion batteries associate with the insufficient energy density, poor stability and safety issues, and the LiCoO2 remains the major cathode that has applied in 3C electronics.

Response to (2): yes it does, There are many items for a LIBs that must be considered, such as insufficient energy density, poor stability and safety issues, economic issues,  simple manufacturing , and so on. But all of these items can not to be put together at once, moreover there are many different between LIBs that used for a laptop or cellphone compared with vehicles. In these paper we focused for decreasing the price and toxic of LIBs based on decreasing the amount of LiCoO2 in viewpoint of both toxic and economic  

 (3): The paper is not well-prepared, as many of tests are involved without any results rather than the discussion.

Response to (3): I revised whole of the manuscript again 

Reviewer 2 Report

This manuscript by Li et al reports the cathode composite material synthesis of various olivine structure materials with partially replacing Fe with Mn, Ti, Co to obtain better battery performances such as cyclability, voltage, thermal stability, etc. 28 samples of ternary composites were synthesized by sol-gel method and characterized using XRD, XPS, and SEM. However, this manuscript still requires much of modification for the publication.

Detailed comments:

1. Authors missed to cite literature for multiple places.

2. Experimental conditions have not been clearly defined. For example, how authors determine cyclability and capacity shown in table 4? 

3. Many typos throughout the text and captions (Ni vs Ti, SiO4 vs PO4, etc.)

4. Figure 9 are quite difficult to understand. Voltage profiles vs capacity should be relatively smooth curves unless the voltages were measured manually. 

Author Response

Dear Editor

Many thanks for your cooperation and referee comments which are pretty helpful for improving the manuscript. In the following there are the responses to the reviewer (point by point)

                                                 Reviewer2#

This manuscript by Li et al reports the cathode composite material synthesis of various olivine structure materials with partially replacing Fe with Mn, Ti, Co to obtain better battery performances such as cyclability, voltage, thermal stability, etc. 28 samples of ternary composites were synthesized by sol-gel method and characterized using XRD, XPS, and SEM. However, this manuscript still requires much of modification for the publication

(1): Authors missed to cite literature for multiple places

Response to (1): I added   more citation in related places

(2) Experimental conditions have not been clearly defined. For example, how authors determine cyclability and capacity shown in table 4? 

Response to (2): Some more explanation has been added in the 3.3 section

(3): Many typos throughout the text and captions (Ni vs Ti, SiO4 vs PO4, etc.)

Response to (3): The typos has been corrected

(4): Figure 9 are quite difficult to understand. Voltage profiles vs capacity should be relatively smooth curves unless the voltages were measured manually.

 Response to (4): Yes due to a large testing for 28 samples and several time for each sample, we measured both manually and standard. Moreover, In an additional Figure (Figure10) including 14, 23, 10, 28, 2, 8, 26 and 22 samples are added to the manuscript where the data have been calculated manually

Reviewer 3 Report

This is just a start of work. The obtained results particularly cycling with relatively low C rate do not show encouroiging results which proves validity of the concept. Much more experiment with different C rates must be perform . Demonstration when the capacily  loss is below 80% of origina ccapacity is the must. The paper should be rejected in the currrent form. 

Author Response

Dear Editor

Many thanks for your cooperation and referee comments which are pretty helpful for improving the manuscript. In the following there are the responses to the reviewer (point by point)

Reviewer3#

This is just a start of work. The obtained results particularly cycling with relatively low C rate do not show encouroiging results which proves validity of the concept. Much more experiment with different C rates must be perform . Demonstration when the capacily  loss is below 80% of origina ccapacity is the must. 

Answer: Although I believe that the data are accurate in this work (as possible) , but due to testing 28 samples and each sample several times, therefor demonstration when the capacity loss is below high percentage  or high C rate for each test takes huge time.. In this stage we have focused to find the best composition, relatively and after this in another work we intend to optimize the efficiency of this composite (Li1.501Co0.389Ti0.055Fe0.055Mn0.501PO4 )in viewpoints of cycling stability, rate performance, high-rate stability, columbic efficiency, impedance and so forth.

Many Thanks for your cooperation and the referee’s comments

Best Regards

Majid Monajjemi

Round 2

Reviewer 1 Report

(1) When discussing the develoopment of LIBs, more trend and representative outcomes of LIBs should be added and cited into the background, such as Nature Energy 2022 DOI:10.1038/s41560-021-00962-y; Advanced Functional Materials 2021, DOI:10.1002/adfm.202107764; Advanced Energy Materials 2022, DOI:10.1002/aenm.202200368; Energy & Environmental Science 2022, DOI:10.1039/D1EE02929K. 

(2) These outcomes aforementioned associate with the solid-state lithium metal batteries or liquid electrolyte lithium metal batteries, which are mainly focusimg on solving the energy density limits of LIBs. However, your article focuses on the cost and toxicity of cathode materials, which should be properly introduced and addressed in the background. 

Author Response

Dear Editor

Many thanks for your cooperation and referee comments which are pretty helpful for improving the manuscript. In the following there are the responses to the reviewer (point by point)

                                                 reviewer1#

Yes

Can be improved

Must be improved

Not applicable

Does the introduction provide sufficient background and include all relevant references?

( )

(x)

( )

( )

Are all the cited references relevant to the research?

(x)

( )

( )

( )

Is the research design appropriate?

(x)

( )

( )

( )

Are the methods adequately described?

(x)

( )

( )

( )

Are the results clearly presented?

(x)

( )

( )

( )

Are the conclusions supported by the results?

(x)

( )

( )

( )

Comments and Suggestions for Authors

(1) When discussing the develoopment of LIBs, more trend and representative outcomes of LIBs should be added and cited into the background, such as Nature Energy 2022 DOI:10.1038/s41560-021-00962-y; Advanced Functional Materials 2021, DOI:10.1002/adfm.202107764; Advanced Energy Materials 2022, DOI:10.1002/aenm.202200368; Energy & Environmental Science 2022, DOI:10.1039/D1EE02929K.

Answer: a paragraph including three above mentioned references have been added (References, 19-21)

(2) These outcomes aforementioned associate with the solid-state lithium metal batteries or liquid electrolyte lithium metal batteries, which are mainly focusimg on solving the energy density limits of LIBs. However, your article focuses on the cost and toxicity of cathode materials, which should be properly introduced and addressed in the background. 

Answer: a paragraph related to the solid-state lithium metal batteries or liquid electrolyte lithium metal batteries has been added and highlighted

                                                 Reviewer2#

Open Review

English language and style

( ) English very difficult to understand/incomprehensible
( ) Extensive editing of English language and style required
(x) Moderate English changes required
( ) English language and style are fine/minor spell check required
( ) I don't feel qualified to judge about the English language and style

Yes

Can be improved

Must be improved

Not applicable

Does the introduction provide sufficient background and include all relevant references?

(x)

( )

( )

( )

Are all the cited references relevant to the research?

(x)

( )

( )

( )

Is the research design appropriate?

(x)

( )

( )

( )

Are the methods adequately described?

(x)

( )

( )

( )

Are the results clearly presented?

(x)

( )

( )

( )

Are the conclusions supported by the results?

(x)

( )

( )

( )

Comments and Suggestions for Authors

I would recommend that this manuscript still need to improve English before publication. 

Answer: Thanks of the positive comments; In addition I agree to improve the English by expert person from molecules team

Reviewer3#

Yes

Can be improved

Must be improved

Not applicable

Does the introduction provide sufficient background and include all relevant references?

( )

(x)

( )

( )

Are all the cited references relevant to the research?

(x)

( )

( )

( )

Is the research design appropriate?

( )

(x)

( )

( )

Are the methods adequately described?

( )

(x)

( )

( )

Are the results clearly presented?

( )

(x)

( )

( )

Are the conclusions supported by the results?

( )

(x)

( )

( )

Comments and Suggestions for Authors

I do not find any new information compared to the former version and my suggestions were ignored.

So I leave the final decision to editors. 

Answer: I had honestly answered to the referee comments in the first round of reviewing about the complexity of testing 28 samples. It is notable for these samples we reached to relative results that by distinguished that this composite (Li1.501Co0.389Ti0.055Fe0.055Mn0.501PO4 ) can be selected as the best one compared with other 27 samples. In addition in future I or other researchers can be improved this samples in view point of the cycling stability, rate performance, high-rate stability, columbic efficiency, impedance and so forth

Many Thanks for your cooperation and the referee’s comments

Best Regards

Majid Monajjemi

Reviewer 2 Report

I would recommend that this manuscript still need to improve English before publication. 

Author Response

(The authors gave the same response as above.)

Reviewer 3 Report

I do not find any new information compared to the former version and my suggestions were ignored. So I eave the final decision to editors. 

Author Response

(The authors gave the same response as above.)
